# Control of Brushless Direct-Current Motors Using Bioelectric EMG Signals

**DOI:** 10.3390/s22186829

**Published:** 2022-09-09

**Authors:** Sebastian Glowinski, Sebastian Pecolt, Andrzej Błażejewski, Bartłomiej Młyński

**Affiliations:** 1Slupsk Pomeranian Academy, Institute of Health Sciences, Westerplatte 64, 76200 Slupsk, Poland; 2The State Higher School of Vocational Education in Koszalin, Lesna 1, 75582 Koszalin, Poland; 3Department of Mechanical Engineering, Koszalin University of Technology, Sniadeckich 2, 75453 Koszalin, Poland

**Keywords:** bioelectric signals, engine control, brushless direct-current motors

## Abstract

(1) Background: The purpose of this study was to evaluate the analysis of measurements of bioelectric signals obtained from electromyographic sensors. A system that controls the speed and direction of rotation of a brushless DC motor (BLDC) was developed; (2) Methods: The system was designed and constructed for the acquisition and processing of differential muscle signals. Basic information for the development of the EMG signal processing system was also provided. A controller system implementing the algorithm necessary to control the speed and direction of rotation of the drive rotor was proposed; (3) Results: Using two muscle groups (biceps brachii and triceps), it was possible to control the direction and speed of rotation of the drive unit. The control system changed the rotational speed of the brushless motor with a delay of about 0.5 s in relation to the registered EMG signal amplitude change; (4) Conclusions: The prepared system meets all the design assumptions. In addition, it is scalable and allows users to adjust the signal level. Our designed system can be implemented for rehabilitation, and in exoskeletons or prostheses.

## 1. Introduction

The human body has created a large area of research, not just in the medical industry. Technology is constantly evolving and introducing new solutions in the field of biomechanics [1]. Owing to the dynamic miniaturization of electronic systems, modern research by doctors and engineers has been widely used in medicine. Biosignals also play a number of particularly important roles. One of these is the use of signals generated by skin tissues that are not integral parts of muscles; that is, sources of the executive systems [2,3]. They can be complex limb prostheses or cooperating devices that support the strength of human muscles [4,5,6]. Basic bioelectric signals include electrocardiograms, electroencephalograms, electromyograms, and electrogastrograms (EGGs). Electrographic signals are used to analyze muscle activity [7,8].

Electromyography (EMG) studies muscle activity by reading the electrical signals generated by muscles [9,10]. Skeletal muscles are responsible for locomotor activity in humans. They set bones in motion and allow people to move and perform daily activities. We consciously controlled a large portion of the muscles. Muscle contractions and stretches respond to the electrical potential generated by the nervous system [10]. EMG signals can be used to diagnose and treat neurological disorders. The use of radiotelemetry in EMG probes has led to its widespread use in professional athletes. For example, in basketball players, it is used when teaching movement patterns, among other uses. As a result, the correlation between the time of muscle activity, strength of the EMG signal, and accuracy of the throw increased [11]. EMG signals are also used in physiotherapy. Meanwhile, in sports, feedback from a patient’s muscle neurons (EMG feedback) is used. They aid in the correct performance of exercises [12]. Patients with paresis are able to see the reaction of the system in an attempt to move the limb. Often, extensive therapeutic devices use data from other sensors, such as accelerometers, strain gauges, or markers, to reconstruct the displacement. As a result, the patient and therapist receive a full reflection of the progress made and the activities that need improvement [13]. Another area of use for muscle signals is assisted exoskeletons, which are designed to support a user’s muscle strength. In addition to being used in the rehabilitation of disabled people, they also provide a solution to supporting healthy people; for example, as an aid in medical care in industrial and military applications [14,15].

Brushless direct-current (BLDC) motors consist of a stationary stator with a winding on it [16,17]. They can be divided into single, two, and three phases. The efficiency of the engine depends on the coil arrangement. To optimize this, the coils should divide the stator disk into equal parts: for a three-phase motor, the stator disk should be divided into three angles of 120° each. Motors can be classified based on the presence of rotor-position sensors in relation to the stator. For this task, Hall sensors are often used. Owing to the Hall effect, they can precisely determine the moment at which subsequent windings should be energized to obtain a smooth rotational motion. In their absence, the electronic system responsible for switching the successive phases must read the rotor position. This phenomenon is called the back electromotive force (BEMF), which induces a voltage in the motor coil as a result of the rotation of the rotor [18,19].

BLDC motors require multiphase power. Because of the lack of a commutator, they require an electronic commutation system. To implement the control, a multiplied H-bridge structure was used, the repetitions of which depended on the number of motor windings. AVR microcontrollers can be used to control the motors [20]. These are programmable integrated circuits. The construction of these devices, as developed by Atmel, was based on the principles of modified Harvard architecture. The memory has been divided into data and commands that use common data and address buses [21,22]. This solution enables an easy data transfer between separate memories. The described microprocessors have a high computing performance and are based on 32 eight-bit registers.

The myoelectric interfaces are used in rehabilitation technology, assistance, and as an input device. A guide of recommendations for myoelectric signal processing according to the application of the interface was presented in [23]. An unambiguous terminology and a good understanding of the nature of the control problem is important for efficient research and communication concerning advanced prosthetic systems [24,25]. The development, advantages, and disadvantages of prosthetic control systems that are current barriers for the transition from laboratory to clinical practice were also discussed by Roche et al. [26]. A comparison of pattern recognition and direct control in eight transhumeral amputees who had TMR in a balanced randomized cross-over study was described by Hargrove [27]. Authors have demonstrated that pattern recognition is a viable option and has functional advantages over direct control. A novel electromyography (EMG)-driven hand exoskeleton for the bilateral rehabilitation of grasping in stroke patients was discussed in [28].

The main purpose of this study is to build and test a system that controls the speed and direction of rotation of a brushless DC (BLDC) motor based on bioelectric EMG signals. The measurement system was characterized in the first part of the work. The system design and the way in which the individual components of the system were used for the acquisition and processing of differential muscle signals were selected. The EMG signal processing method was then characterized. By using two muscle groups (biceps brachii and triceps), it was possible to control the direction and speed of rotation of the drive unit. Finally, a brief discussion and the limitations of the study is presented. Diagrams of the proposed solutions are presented as Appendix A for this manuscript.

## 2. Materials and Methods

### 2.1. Measuring System

A general scheme of the measurement system is shown in Figure 1a. The operation of the system began by collecting the voltage data from the skin surface. These EMG signals are generated by motor units found in the muscles. The collected data had a low amplitude (approx. 1–2 mV); therefore, a high gain was initially applied. This allowed us to separate the relevant data from the disturbance. The signal was then cleaned of unwanted frequencies using a high-pass filter and straightened using a full-wave rectifier. Finally, the signal was smoothed using a low-pass filter and converted into a digital value. The obtained values were processed by a processing unit (CPU), coded, and sent to the controller via the Bluetooth interface.

In the executive system, the tasks of servicing communication and controlling the engine rotation were divided into two separate computing units. First, it received an encoded message from a measurement element. This was processed and sent to the controller to control the engine operation. The control part generated six signals that control the switching of the successive pairs of keying transistors. For speed control, three of the above-mentioned signals were PWM signals. The filling depended on the measured values. These signals went to gate drivers that directly switched on the MOSFETs. For security purposes, a code that prohibits starting the engine without calibrating the input signal range was implemented. To confirm the start and end of signal calibration, a button marked as “calibration” was installed (Figure 1b).

To verify the correctness of the operation, a computer measuring station was used in which the signals were recorded using a National Instruments measurement card (model NI USB-6211) (Figure 2f) [29]. The tests used analog inputs, which were characterized by a 16-bit resolution with a measurement range of ±10 V and a maximum data acquisition speed of 250 kS/s. To record the information, the LabView environment of National Instruments was used, in which a short program was created to support communication with the measurement card and save the collected data to a file. In the study, reusable suction-cup probes were used (Figure 2e). Owing to the short duration of the signal which determines the rotation of the motor, the Arduino prototype platform was used to convert it into a PWM signal. Subsequently, low-pass filtration with a cutoff frequency of 7.958 Hz was used, which contributed to the acquisition of an analog filter with an amplitude of 0–5 V, determining a speed of 0–9108 rpm in accordance with the engine parameters when powered at 9.9 V.

### 2.2. Measurement Method

The EMG signal measurement began by collecting myoelectric signals from the surface of the skin at the active muscle site. For this purpose, suction-cup Ag/AgCl probes were used [30]. To improve contact and reduce the impact of skin impedance at the contact points, it was degreased, and a gel was used to increase the adhesion of the probes used. The first step in analog signal processing was to amplify the collected data. The goal was to isolate the signal from the background noise. An AD620ARZ instrumental amplifier was used for the amplification circuit [31]. It is characterized by a wide gain capacity (up to 10,000 times), a low input disequilibrium voltage (50 μV), and a low drift level (0.6 μV/°C). An external resistor *R_g_* was chosen to specify the differential gain of the input signal (Figure 3a). Assuming that the initial gain of the differential signal is *G* = 500, the following relationship can be obtained for the required resistor value:(1)$$R_{g}=\frac{49.9\;k\Omega }{G-1}=\frac{49.9\;k\Omega }{500-1}\approx 0.099\;k\Omega$$
where:*R_g_* is the resistance value of the calibration resistor [Ω],*G* is the circuit amplification required.

The equal gain was obtained using the closest available resistor (100Ω):(2)$$G=\frac{49.4}{0.1}+1=495$$

To verify the operation of the system, a test was conducted using a measuring stand equipped with a National Instruments USB-6211 measuring card with symmetrical analog inputs, and a measurement range of ±10 V. The signals from the biceps brachii were recorded at a sampling rate of 10 kHz. The recorded signal for a single contraction is shown in Figure 3b. The signal amplitude was approximately 2 V which, when divided by the generated gain (495), provided a range of changes in the source signal of 4 mV. This value is consistent with the literature data (0.05–10 mV) [32,33]. In the next stage, the frequency of the tested signal was checked. A fast Fourier transform was used in MATLAB (USA) [34]. Most of the signal bandwidth was in the range up to 500 Hz.

An inverting amplifier with a gain of *G* = 2 was used for the initial amplification. A quad LM324 amplifier was used for the DIP14 housing. Consequently, four operational amplifiers were obtained, which served as the subsequent stages of signal processing in one integrated circuit (Figure 4a). One of these was used in the inverting amplifier configuration.

The values of resistors *R*_1_ and *R*_2_ were determined based on their dependence on the inverting amplifier gain. Assuming that the value of the resistance *R*_1_ is 10 kOhm and the value of the gain *G* = 2, the resistance *R*_2_ was calculated as:(3)$$G=\frac{U_{out}}{U_{in}}=\frac{R_{2}}{R_{1}}\Rightarrow R_{2}=R_{1}\cdot G=10\;k\Omega \cdot 2=20\;k\Omega$$
where:*U_out_* is the output voltage of the inverting amplifier circuit [V],*U_in_* is the input voltage of the inverting amplifier circuit [V].

By using the circuit (Figure 4a) and feeding the output of the instrumental amplifier to the input of the inverting amplifier, a signal with a total gain of *G* = −990 (Figure 4b).

High-pass filtration was performed in the next stage. The purpose of this filtering was to transform the input analog signal and isolate the low-frequency signal. This allows for further processing of the amplitude-cleaned signal with a frequency lower than the cutoff of the designed filter, it and eliminates the DC shift that occurred in the previous stage. Therefore, a first-order active filter without amplification was developed (Figure 5a). Another sub-structure of the LM324 amplifier was used during pre-amplification. The cutoff frequency was assumed to be approximately 100 Hz. After analyzing the rows of capacitors available on the market, it was decided to use a capacitor with a capacity of 0.01 µF, which was the basis for determining the value of the resistor:(4)$$R=\frac{1}{2\pi CF_{g}}=\frac{1}{2\cdot 3.14\cdot \left(0.01\cdot 10^{-6}\right)\cdot 100}\approx 150\;k\Omega$$
where:*Fg* is the cutoff frequency of the first-order high-pass filter [Hz],*R* is the resistance of the resistor R9 [Ω],*C* is the capacitor capacity C3 [F].

The determined filter can be represented by the following operator transfer function:(5)$$G\left(s\right)=\frac{1}{1+RCs}$$
where:*G*(*s*) is the operator transmittance of the first-order high-pass filter system,*R* is the resistance of the resistor R9 [Ω],*C* is the capacitor capacity C3 [F].

The frequency-phase characteristics of the designed filter were determined using relationship (5) and a MATLAB package (Figure 5b).

After processing the signal with the high-pass filter, the negative amplitude was removed because the microcontroller used at a later stage did not have symmetrical inputs (±12 V), but analog inputs had an acceptable measuring range of 0–5 V. The removal of this amplitude generated data with extremely high dynamics of change, which significantly hindered digital processing. The solution reflected the negative amplitudes relative to the equilibrium position and then smoothened them. Such a procedure yields a signal with significantly reduced dynamics which is, at the same time, locally stable, enabling its unambiguous interpretation by means of a microcontroller. For this purpose, a precise full-wave rectifier was developed, the design of which was based on two operational amplifiers (Figure 6a).

Its operation can be presented in two steps: when the signal V_0_ > 0 and when V_0_ < 0. In the first case, a positive input signal was applied to the inverting input of A1, causing it to invert. Subsequently, it encountered diode D1, which was blocked, and diode D2 in the forward direction. As long as there was no current flow through the resistor R between inputs 13 and 3, both inputs remained equipotential, implying that they have the same potential. Therefore, assuming that input no. 3 has a potential of 0 V, input no.13 also has 0 V at the same time. When V_0_ < 0, module A1 inverted the negative signal in the positive direction, it followed the forward-facing diode D2, omitting elements that did not affect the output signal. This can be represented as a simplified circuit (Figure 6c).

In the next stage, the signal was prepared for free reading using a microcontroller. Low-pass filtration was used to limit the speed of change in the signal to the extent that the signal would change its characteristics from AC to DC. Its steep slopes were softened, which caused the frequency of the signal to change. An active first-order low-pass filter was used, whose structure was based on an operational amplifier and passive elements in the form of resistors and capacitors. The cutoff frequency *F_g_* was assumed to be 2 Hz, which should sufficiently smooth the processed signal with a high EMG frequency. In line with these assumptions, a capacitor with a capacity of 1μF was selected from a series of types as the basis for the calculations, which allowed the determination of the resistance of the system resistor
(6)$$R=\frac{1}{2\pi F_{g}C}=\frac{1}{2\cdot 3.14\cdot 2\cdot 1\cdot 10^{-6}}\approx 79.62\;k\Omega$$

In order to obtain the determined resistance, the closest values of the resistors were selected and connected in series. The replacement resistance of the selected elements was 81.7 kΩ, which translates to a cut-off frequency of 1.95 Hz. Using the designed filter, a test was carried out to check its filtration capabilities (Figure 7). A signal processed using a precise full-wave rectifier was used as the signal source. The received signal was smoothed out but inverted, and its amplitude was too low to be easily processed by the microcontroller. To improve its properties, a final inverting amplifier with adjustable gain was fabricated. Using this system, it was possible to adjust the amplitude of the output signal, according to the individual physical conditions of the user. This circuit was similar to that used in the previous sections of the LM324 operational amplifier. A 1 kΩ resistor was used as the basis for the gain in combination with a 20 kΩ precision potentiometer in the feedback branch. Assuming that we do not lower the potentiometer resistance below 1 kΩ, this gives a gain adjustment in the range of 0–20 times. A schematic of the system is shown in Figure 8a.

Amplified and inverted signals were obtained for digital processing (Figure 8b). For this purpose, an analog-to-digital converter was used. It was assumed that, to adequately reproduce individual levels of input data, a system with a minimum resolution of 12 bits should be used, which yields a signal change of 1.2 mV. Therefore, we decided to use the MICROCHIP MCP3221 converter in the SOT-23-5 housing, which satisfied the above assumptions [35]. Moreover, the converter had a digital TWI communication port for data transmission and enabled the connection of up to eight devices in a single communication bus [36]. Therefore, in the future, the system can be expanded with further measurement loops for the signal analysis of other muscle groups.

For signal processing, a microcontroller was used to collect data from the two muscle groups and prepare the data for further wireless communication. An ATMEGA328 system was used [37]. The implemented program received analog signals in the form of digital data packets, processed them, transmitted them to the wireless transmission module, and communicated with the paired circuit of the BLDC motor controller.

## 3. Results

This study was conducted in two independent stages. In the first stage, the operation of a single sEMG analog signal processing board was verified. For this purpose, two probes were placed on the bicep of the examiner and a third (reference) probe was placed near the elbow. The person conducting the examination performed three quick tensions of the biceps brachial muscle within 10 s, and a measurement card was used to record the data for further analysis. The next step was to verify the correct operation of the entire system. The main assumption of the project was to control the direction and speed of rotation of the BLDC motor based on EMG signals. To verify the response to the set signal coming from the potential generated by the muscle strength, two muscle groups were used: the biceps and triceps of the arm, owing to their antagonistic effect. This allowed us to verify the correct change in the direction and speed of operation. The signals obtained after the analog processing were recorded for verification. During the examination, the biceps muscle was tightened by accelerating the motor, followed by a loosening and shortening of the triceps by accelerating the motor in the other direction.

### 3.1. Examination of the EMG Analog Data Processing System

The performance of the system was analyzed according to the presented research methodology. Figure 9a shows the raw signal of the three muscle tensions after the initial amplification with an instrumental enhancer. It shows the shift of the signal from zero to a level of 1 V, and a maximum amplitude of approximately 2.5 V. The signal exhibited a very high frequency of change. To use it as a control signal, it must be processed further. Figure 9a below shows the frequency spectrum of the first waveform. It can be concluded that most of the bands were in a range of up to 1000 Hz. The second stage amplified and inverted the signal, which caused a shift from zero to the −2 V level. The peak-to-peak voltage increased to approximately 5 V. In the plot, the range that covers most of the signal band did not change as a function of frequency. On the other hand, its amplitudes were changed, reaching about 0.04 V at their peak (Figure 9b).

Figure 10a shows the signals obtained after high-pass filtration. The DC offset is noticeable. The highest amplitude of the upper signal reached a value of approximately 3 V, and the lower one reached approximately 0.018 V. The signal still required processing to perform digital processing. The obtained waveforms show the signals after processing with a full-wave rectifier (Figure 10b). The upper graph shows that the negative values were reflected against zero, the waveform took the form of three excitations with amplitudes in the range of 0–1.8 V, and the high-frequency change within each of them. The lower graph shows a significant decrease in amplitude in relation to the waveform from the third stage and a change in the distribution of the signal band.

A low-pass filter was used, the purpose of which was to rotate the processed signal and remove high frequencies. As a result, three signals were obtained with a maximum amplitude of up to 0.4 V and low noise. The frequency spectrum of the signal shows the removal of high frequencies, and most of its bandwidth was in a range of up to 20 Hz (Figure 11a). Figure 11b shows the sEMG signals for three muscle tensions. The maximum amplitude was approximately 4.5 V. The visible waveform had no noise, and its changes were gentle, clearly showing a change in the potential recorded by the data acquisition system.

By analyzing the obtained results, it can be stated that the analog processing of the read EMG signal was correct. The final signal obtained can be interpreted using a digital processing system, which provides the basis for the second stage of this study.

### 3.2. Test of the BLDC Motor Rotation Speed and Direction Control System Based on the EMG Signal

The study of the BLDC motor rotation speed and direction control system were analyzed using a measuring stand and processed in MATLAB (Figure 12). During the tests, the engine was rotated at a maximum speed of 8355 rpm, which was recorded using a Testo model 470 tachometer [38]. During this period (approximately 4 s), the amplitude of the corresponding signal (red) reached a level of about 3.3 V. The speed characteristics in relation to the signal amplitude were linear, which yields the determination of the engine speed at any time during the test. Although the graphs of the EMG signals from both muscles overlap, the applied algorithm transfers control over the control to the group that crosses the minimum threshold first. The obtained waveforms show that in the first case, the motor followed the change in the amplitude of the signal from the biceps (black), whereas in the second, it followed that from the triceps (blue). This is indicated by the signals in the rotational direction (green and blue).

By analyzing the obtained time courses, it can be concluded that the engine response was delayed in relation to the muscle shortening signal by about 0.5 s, which may be caused by the delay generated during communication, or too little optimization of the source code. The graphs show the correct response of the control system, both to a change in the tense muscle, resulting in a change in the rotational direction, and to the correction of the BLDC motor rotational speed, depending on the level of the bioelectric EMG signals, which proves the correct operation of the system.

In general, the operation of the system can be presented in following steps. First, the processed analog signal obtained from the muscle is converted into digital form by the microcontroller (CPU). Next, the digital signal is encoded and compressed into a message, which is sent via a Bluetooth interface to a second microcontroller (CPU) that controls the engine. Here, the message is decoded and then processed to generate six signals controlling the switching of successive pairs of MOSFET transistors. On this basis, the process of controlling the speed and direction of the BLDC motor is implemented. As one can see, the complex process of processing the raw EMG signal, transmission via the wireless path, and the decoding process directly affects the delay in motor response. This can be seen in the plot of the correlation function of processed EMG signals and motor velocity (Figure 12). It is possible to optimize the response speed of the system; however, this was not the main objective of current task. On the other hand, the developed system can be used in prosthetic systems to control the grip or rotation of a prosthetic hand, where the recorded delay does not negatively affect the functionality of the device and, in fact, is sometimes desirable, due to the perception of the user.

## 4. Discussion

The developed data acquisition system correctly received and processed the EMG signals. The received signal was processed using digital circuits, and the frequency of the input signals was consistent with parameters reported in the literature. It was confirmed that the parameters of the individual filters were selected correctly. This made it possible to obtain the intended results at the individual stages of source-signal processing. The processed signal was successfully transmitted wirelessly to the control system. The control algorithm successfully controlled the actuator system while maintaining satisfactory dynamics of the BLDC motor speed changes. The control system changed the rotational speed of the brushless motor with a delay of about 0.5 s in relation to the registered EMG signal amplitude change. By using two muscle groups, the biceps brachii and triceps, it was possible to control the direction and speed of rotation of the driving unit. The prepared system meets all the design assumptions. In addition, it is scalable and allows users to adjust their signal level. The engine speed is scalable and related to the amplitude of the recorded EMG signal.

During the course of the experiment, the amplitude of the signal from Channel 1 (biceps brachii) was significantly higher than that from Channel 2 (triceps). Fewer body fat may be the cause, requiring further research. Because of the type of motor used, it cannot be started smoothly. At takeoff, the engine pursues a minimum speed of approximately 10% of its maximum speed. In future, the value of the EMG signal should be assessed as a function of load on the forearm.

## Figures and Tables

**Figure 1 sensors-22-06829-f001:**
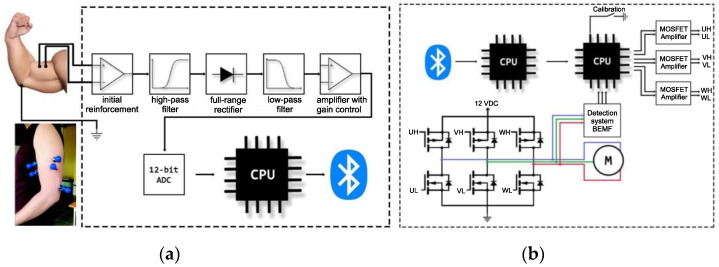
General scheme of the measurement system construction (**a**), diagram of the actuating system (**b**).

**Figure 2 sensors-22-06829-f002:**
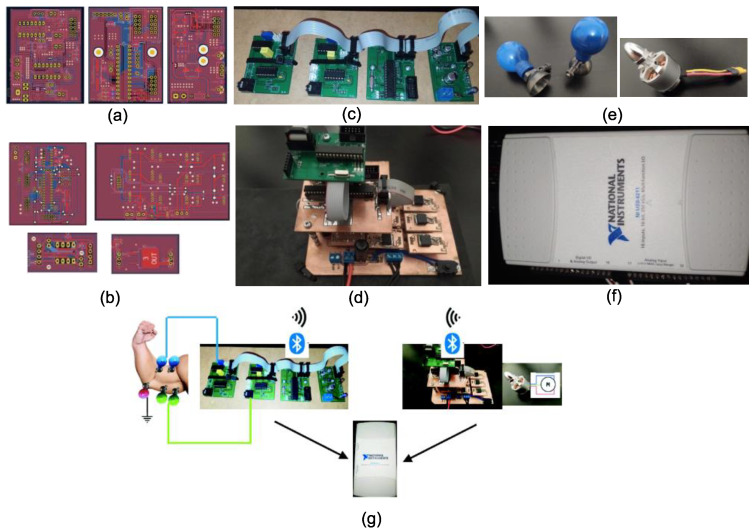
Design of printed circuit boards of data acquisition circuit (KiCad) (**a**), design of PCBs of BLDC motor controller (**b**), ready system of data acquisition without Bluetooth communication module (**c**), ready circuit of BLDC controller without connected Bluetooth communication module (**d**), suction-cup probes measuring and BLDC motor without integrated Hall sensors, (**e**) measuring card by National Instruments model NI USB-6211, (**f**) schematic diagram of the connection state or rest state of the entire system (**g**).

**Figure 3 sensors-22-06829-f003:**
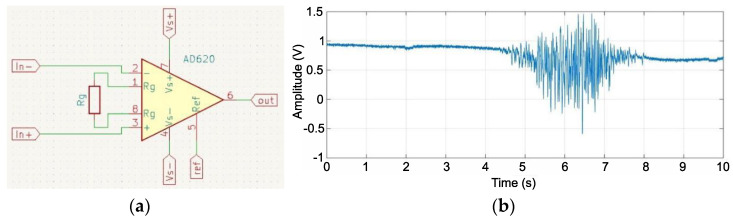
Scheme of connection of the instrumental amplifier circuit model AD620 (**a**), EMG biceps brachii signal amplified 495 times by the instrumental amplifier AD620ARZ (**b**).

**Figure 4 sensors-22-06829-f004:**
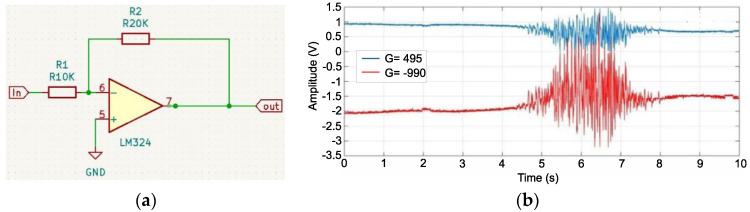
Connection diagram of one of the LM324 operational amplifier circuits in the inverting amplifier configuration (**a**), comparison diagram of the signal with the gain G = 495 (blue) and *G* = −990 (red) (**b**).

**Figure 5 sensors-22-06829-f005:**
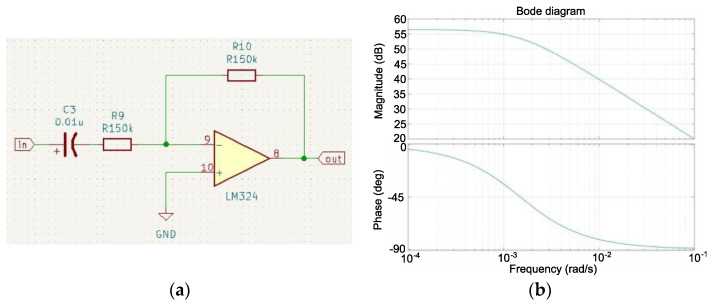
Connection diagram of the second LM324 operational amplifier circuit in the configuration of the first-order high-pass filter (**a**), frequency-phase characteristic of the high-pass filter (**b**).

**Figure 6 sensors-22-06829-f006:**
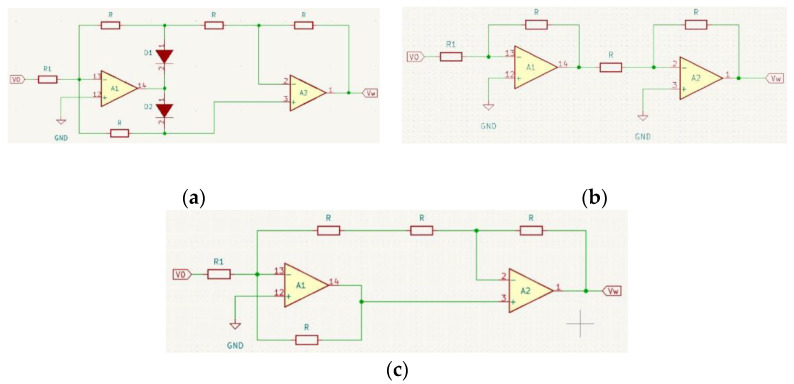
Diagram of a precise full-wave amplifier (**a**), equivalent circuit for simulation of a full-wave rectifier for condition V_0_ > 0 (**b**), equivalent circuit for simulation of a full-wave rectifier for the condition V_0_ < 0 (**c**).

**Figure 7 sensors-22-06829-f007:**
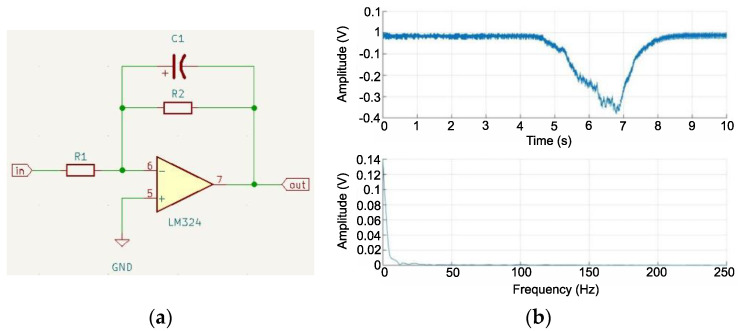
Connection diagram of LM324 operational amplifier in configuration of the first-order low-pass filter (**a**), time and frequency diagram of signal after low-pass filtering (**b**).

**Figure 8 sensors-22-06829-f008:**
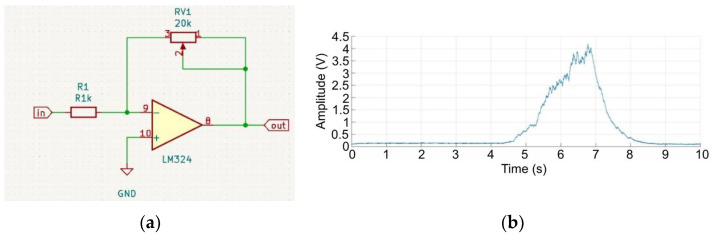
Connection diagram of one of the LM324 operational amplifier circuits in an inverting amplifier configuration with adjustable gain (**a**), time and frequency diagram of the signal after low-pass filtering (**b**).

**Figure 9 sensors-22-06829-f009:**
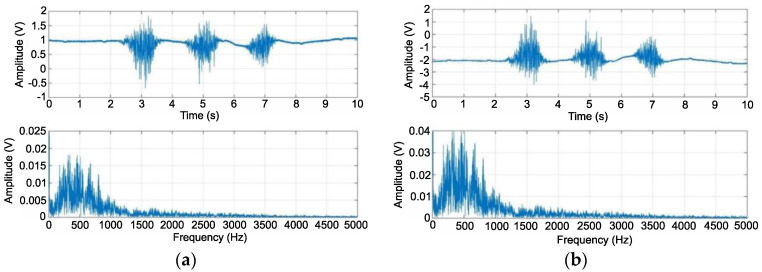
EMG signal as a function of time and frequency after amplification using instrument amplifier (**a**), EMG signal as a function of time and frequency after amplification using inverting amplifier (**b**).

**Figure 10 sensors-22-06829-f010:**
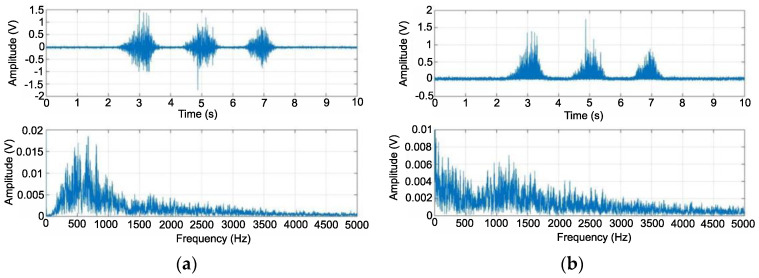
EMG signal as a function of time and frequency after high-pass filtering (**a**), EMG signal as a function of time and frequency after full-wave rectification (**b**).

**Figure 11 sensors-22-06829-f011:**
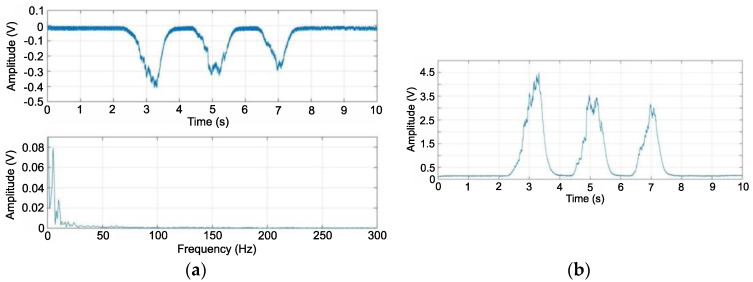
EMG signal as a function of time and frequency after low-pass filtering (**a**), EMG signal as a function of time after amplification using an inverting amplifier with adjustable gain (**b**).

**Figure 12 sensors-22-06829-f012:**
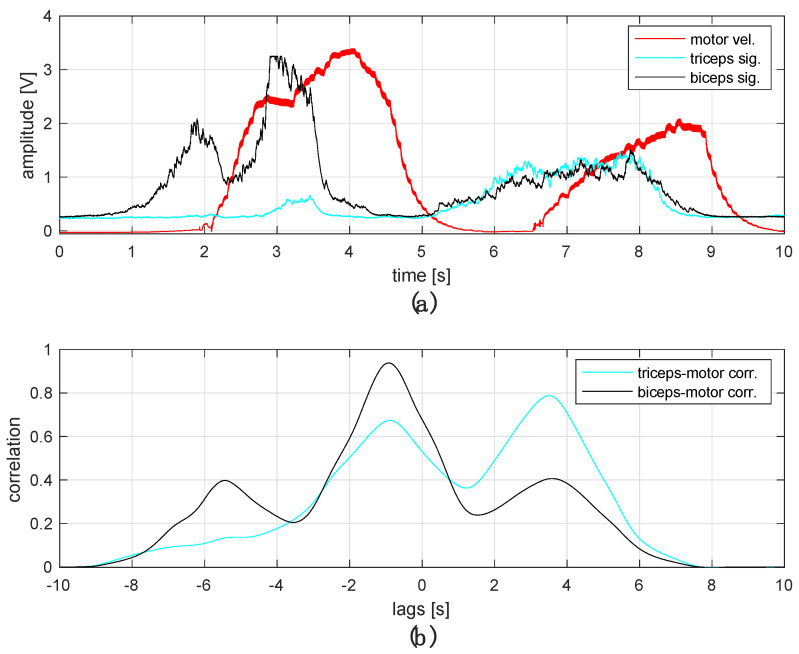
Reaction of the engine to a given EMG control signal (color: black—biceps signal; blue—triceps signal; red—analog interpretation of engine; speed, green—first direction of rotation, blue—second direction of rotation) (**a**), normalized correlation between the processed triceps and biceps signals and motor reactions (**b**).

## Data Availability

The datasets used and/or analyzed during the current study are available from the coauthor (sebastian.pecolt@tu.koszalin.pl) upon reasonable request.

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
