# Peer review of "Control of Brushless Direct-Current Motors Using Bioelectric EMG Signals"

_sensors, 2022, doi:10.3390/s22186829_

Round 1

Reviewer 1 Report

This paper presents a system for controlling the speed and direction of a brushless DC motor using bioelectrical signals. By combining bioelectrical signal analysis and motor control circuit design, motor control can be achieved through the biceps and triceps muscles. This study is a preliminary study and further research is needed.

(1) The author should give a schematic diagram of the tester with the electrodes attached to the body.

(2)The author should give a schematic diagram of the connection state or test state of the entire system

(3) For EMG signals, the authors should give a comparison of the original signal and the processed signal at each stage

(4)The author should explain in more detail the following effect of the motor on the EMG signal

Author Response

Dear Reviewer,

responses are included in a file: Reviewer 1_response

Reviewer 2 Report

Instead of the first reference [1] (self-cite), a more thorough overview of the related work in the field of Myoelectric Control Systems discussing more relevant references of academic quality are needed, preferably as a separate subsection.

Feels like many sources were neglected, for example, here's a few:

https://doi.org/10.1016/j.bspc.2007.07.009
https://doi.org/10.1109/TNSRE.2012.2196711
https://doi.org/10.3390/s130912431
https://doi.org/10.1007/s40137-013-0044-8
https://doi.org/10.1016/j.bspc.2019.02.011
https://doi.org/10.1038/s41598-017-14386-w
https://doi.org/10.1109/ACCESS.2019.2963881
https://doi.org/10.1109/TOH.2015.2417570

From 30 (rather a small amount) of references used, 23% of them are trivial [22-24, 27-29, 31], so improving literature review aspect is a priority.

Most of the manuscript is dedicated to explaining the hardware and implementation, but evaluation aspect is weak. Main experimental result is 0.5 s delay, evaluated using an eye from a single graph - Figure 12? More such graphs would be useful, as well as a more elegant way of figuring out the average delay through a plot of cross-correlation (maximum correlation would indicate perfect sync) - the sample cross correlation function (CCF) or even the multichannel cross-correlation coefficient (MCCC), see:

https://doi.org/10.1007/1-4020-7769-6_8 http://dx.doi.org/10.1109/TSA.2004.833008

Author Response

Dear Reviewer,

thank you very much for your effort and insightful review of our article. Comments and responses are included in the attached file Reviewer 2_response.

Reviewer 3 Report

The paper contents are quite interesting, and can provide a basis for further research in prosthesis. A few suggestions are given next:

- Is this phrase from line 29 correct? "One is the use of signals generated by tissues that are not integral 29 parts of the body as sources of control for the various executive systems [2,3]."

- Use the ° symbol for degrees on line 58 or the word "degrees".

- Use decimal point in the whole document (e.g., in Equation (1) there is both a decimal point and a decimal comma).

- Some notation can be improved, specially the V0 notation on page 6.

- Checking again the document to improve redundant wording (e.g., the phrase "Using this system, it is possible to adjust the amplitude of the output signal can be adjusted according to the individual physical conditions of the user." on line 246).

- Remove the Acknowledgements sections if it is not required.

Author Response

Dear Reviewer,

thank you very much for your effort and insightful review of our article. The comments are included in the attached file: Reviewer 3_response.

Round 2

Reviewer 1 Report

The reviewer's comments have been well addressed. Thanks.